# Act3D: 3D Feature Field Transformers for Multi-Task Robotic Manipulation

**Theophile Gervet**[*,1]    **Zhou Xian**[*,2]    **Nikolaos Gkanatsios**[2]    **Katerina Fragkiadaki**[1]

[1]Machine Learning Department    [2]Robotics Institute
School of Computer Science
Carnegie Mellon University
{tgervet, xianz1, ngkanats, katef}@cs.cmu.edu

[act3d.github.io](https://act3d.github.io)

**Abstract:** 3D perceptual representations are well suited for robot manipulation as they easily encode occlusions and simplify spatial reasoning. Many manipulation tasks require high spatial precision in end-effector pose prediction, which typically demands high-resolution 3D feature grids that are computationally expensive to process. As a result, most manipulation policies operate directly in 2D, foregoing 3D inductive biases. In this paper, we introduce Act3D, a manipulation policy transformer that represents the robot's workspace using a 3D feature field with adaptive resolutions dependent on the task at hand. The model lifts 2D pre-trained features to 3D using sensed depth, and attends to them to compute features for sampled 3D points. It samples 3D point grids in a coarse to fine manner, featurizes them using relative-position attention, and selects where to focus the next round of point sampling. In this way, it efficiently computes 3D action maps of high spatial resolution. Act3D sets a new state-of-the-art in RL-Bench, an established manipulation benchmark, where it achieves 10% absolute improvement over the previous SOTA 2D multi-view policy on 74 RLBench tasks and 22% absolute improvement with 3x less compute over the previous SOTA 3D policy. We quantify the importance of relative spatial attention, large-scale vision-language pre-trained 2D backbones, and weight tying across coarse-to-fine attentions in ablative experiments. Code and videos are available at our project site: [https://act3d.github.io/](https://act3d.github.io/).

**Keywords:** Learning from Demonstrations, Manipulation, Transformers

## 1 Introduction

Solutions to many robot manipulation tasks can be modeled as a sequence of 6-DoF end-effector poses (3D position and orientation). Many recent methods train neural manipulation policies to predict 3D end-effector pose sequences directly from 2D images using supervision from demonstrations [1, 2, 3, 4, 5, 6]. These methods are typically sample inefficient: they often require many trajectories to handle minor scene changes at test time and cannot easily generalize across camera viewpoints and environments, as mentioned in the respective papers and shown in our experiments.

For a robot policy to generalize under translations, rotations, or camera view changes, it needs to be spatially equivariant [7], that is, to map 3D translations and rotations of the input visual scene to similar 3D translations and rotations for the robot's end-effector. Spatial equivariance requires predicting 3D end-effector locations through 2D or 3D action maps, depending on the action space considered, instead of regressing action locations from holistic scene or image features. Transporter

---

* Equal contribution

7th Conference on Robot Learning (CoRL 2023), Atlanta, USA.

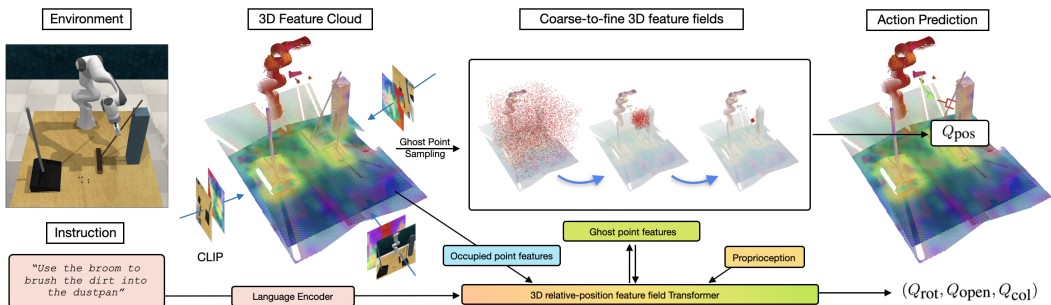

Figure 1: **Act3D** is a language-conditioned robot action transformer that learns 3D scene feature fields of arbitrary spatial resolution via recurrent coarse-to-fine 3D point sampling and featurization using relative-position attentions. Act3D featurizes multi-view RGB images with a pre-trained 2D CLIP backbone and lifts them in 3D using sensed depth. It predicts 3D location of the end-effector using classification of the 3D points of the robot's workspace, which preserves spatial equivariance of the scene to action mapping.

networks [8] introduced a spatial equivariant architecture for 4-DoF robot manipulation: they re-project RGB-D input images to a top-down image and predict robot end-effector 2D translations through a top-down 2D action map. They showed better generalization with fewer training demonstrations than prior works. However, they are limited to top-down 2D worlds and 4-DoF manipulation tasks. This begs the question: how can we extend spatial equivariance in action prediction to general 6-DoF manipulation?

Developing spatially equivariant 6-DOF manipulation policies requires predicting 3D action maps by classifying 3D points in the robot's workspace as candidates for future 3D locations for the robot's end-effector. Predicting high-resolution 3D action maps, necessary for fine-grained manipulation tasks, poses a computational challenge over their 2D counterparts due to the extra spatial dimension. Voxelizing the robot's 3D workspace and featurizing the 3D voxels at high resolution is computationally demanding [9]. The next end-effector pose might be anywhere in free space, which prevents the use of sparse 3D convolutions [10, 11] to selectively featurize only part of the 3D free space. To address this, recent work of PerAct [1] featurizes 3D voxels using the latent set bottlenecked self-attention operation of Perceiver [12], whose complexity is linear to the number of voxels as opposed to quadratic, as the all-to-all self attention operations. However, it gives up on spatial disentanglement of features due to the latent set bottleneck. Other methods avoid featurizing points in 3D free space altogether and instead regress an offset for the robot's 3D locations from a detected 2D image contact point [2, 13, 14], which again does not fully comply with spatial equivariance.

In this paper, we introduce Act3D, a language-conditioned transformer for multi-task 6 DoF robot manipulation that predicts continuous resolution 3D action maps through adaptive 3D spatial computation. Act3D represents the scene as a continuous 3D feature field. It computes a scene-level physical 3D feature cloud by lifting features of 2D foundational models from one or more views using sensed depth. It learns a 3D feature field of arbitrary spatial resolution via recurrent coarse-to-fine 3D point sampling and featurization. At each iteration, the model samples 3D points in the whole workspace and featurizes them using relative spatial cross-attention [15] to the physical 3D feature cloud. Act3D predicts 3D end-effector locations by scoring 3D point features, and then regresses the 3D orientation and opening of the end-effector. At inference time, we can trade-off compute for higher spatial precision and task performance by sampling more 3D points in free space than the model ever saw at training time.

We test Act3D in RLBench [16], an established benchmark for learning diverse robot manipulation policies from demonstrations. We set a new state-of-the-art in the benchmark in both single-task and multi-task settings. Specifically, we achieve a 10% absolute improvement over prior SOTA on the single-task setting introduced by HiveFormer [2] with 74 tasks and a 22% absolute improvement over prior SOTA in the multi-task setting introduced by PerAct [1] with 18 tasks and 249 variations.

We also validate our approach on a Franka Panda with a multi-task agent trained from scratch on 8 real-world tasks with a total of just 100 demonstrations (see Figure 2). In thorough ablations, we show the importance of the design choices of our architecture, specifically, relative spatial attention, large-scale vision-language pre-trained 2D backbones, high resolution featurization and weight tying across coarse-to-fine attentions.

In summary, our contributions are: **1.** A novel neural policy architecture for language-conditioned multi-task 6-DoF manipulation that both reasons directly in 3D and preserves locality of computation in 3D, using iterative coarse-to-fine translation-invariant attention. **2.** Strong empirical results on a range of simulated and real-world tasks, outperforming the previous SOTA 2D and 3D methods on RLBench by large absolute margins, and generalizing well to novel camera placements at test time. **3.** Thorough ablations that quantify the contribution of high-resolution features, tied attention weights, pre-trained 2D features, and relative position attention design choices.

## 2    Related Work

**Learning robot manipulation from demonstrations**    Many recent work train multi-task manipulation policies that leverage Transformer architectures [1, 2, 3, 5, 17, 18] to predict robot actions from video input and language instructions. End-to-end image-to-action policy models, such as RT-1 [5], GATO [18], BC-Z [19], and InstructRL [3], directly predict 6-DoF end-effector poses from 2D video and language inputs. They require many thousands of demonstrations to learn spatial reasoning and generalize to new scene arrangements and environments. Transporter networks [8] and their subsequent variants [20, 21, 22] formulate 4-DoF end-effector pose prediction as pixel classification in 2D overhead images. Thanks to the spatial equivariance of their architecture, their model dramatically increased sample efficiency over previous methods that regress end-effector poses by aggregating global scene features. However, they are limited to top-down 2D planar worlds with simple pick-and-place primitives. 3D policy models of C2F-ARM [4] and PerAct [1] voxelize the robot's workspace and are trained to detect the 3D voxel that contains the next end-effector key-pose. Spatially precise 3D pose prediction requires the 3D voxel grid to be high resolution, which comes at a high computational cost. C2F-ARM [4] uses a coarse-to-fine voxelization to handle computational complexity, while PerAct [1] uses Perceiver's latent bottleneck [12] to avoid voxel-to-voxel self-attention operations. Act3D avoids 3D voxelization altogether and instead represents the scene as a continuous resolution 3D feature field. It samples 3D points in the empty workspace and featurizes them using cross-attentions to the physical 3D point features.

**Feature pre-training for robot manipulation**    Many 2D policy architectures bootstrap learning from demonstrations from frozen or finetuned 2D image backbones [23, 24, 19, 25] to increase experience data sample efficiency. Pretrained vision-language backbones can enable generalization to new instructions, objects, and scenes [26, 21]. In contrast, SOTA 3D policy models are typically trained from scratch from colored point clouds input [1, 4, 27]. Act3D uses CLIP pre-trained 2D backbones [28] to featurize 2D image views and lifts the 2D features in 3D using depth [29, 30]. We show that 2D feature pretraining gives a considerable performance boost over training from scratch.

**Relative attention layers**    Relative attentions have shown improved performance in many 2D visual understanding tasks and language tasks [31, 32]. Rotary embeddings [33] implement relative attention efficiently by casting it as an inner-product in an extended position feature space. In 3D, relative attention is imperative as the coordinate system is arbitrary. 3D relative attentions have been used before in 3D Transformer architectures for object detection and point labelling [34, 35]. We show in Section 4 that relative attentions significantly boost performance of our model.

## 3    3D Feature Field Transformers for Multi-Task Robot Manipulation

The architecture of Act3D is shown in Figure 1. It is a policy transformer that, at a given timestep $t$, predicts a 6-DoF end-effector pose from one or more RGB-D images, a language instruction,

and proprioception information regarding the robot's current end-effector pose. Following prior work [36, 1, 2, 3], instead of predicting an end-effector pose at each timestep, we extract a set of *keyposes* that capture bottleneck end-effector poses in a demonstration. A pose is a keypose if (1) the end-effector changes state (something is grasped or released) or (2) velocities approach near zero (a common occurrence when entering pre-grasp poses or entering a new phase of a task). The prediction problem then boils down to predicting the next (best) keypose action given the current observation. At inference time, Act3D iteratively predicts the next best keypose and reaches it with a sampling-based motion planner, following previous works [1, 2].

We assume access to a dataset of $n$ demonstration trajectories. Each demonstration is a sequence of observations $O = \{o_1, o_2, .., o_t\}$ paired with continuous actions $A = \{a_1, a_2, .., a_t\}$ and, optionally, a language instruction $l$ that describes the task. Each observation $o_t$ consists of RGB-D images from one or more camera views; more details are in Appendix 7.2. An action $a_t$ consists of the 3D position and 3D orientation (represented as a quaternion) of the robot's end-effector, its binary open or closed state, and whether the motion planner needs to avoid collisions to reach the pose:

$$a = \{a_{\text{pos}} \in \mathbb{R}^3, a_{\text{rot}} \in \mathbb{H}, a_{\text{open}} \in \{0, 1\}, a_{\text{col}} \in \{0, 1\}\}$$

Next, we describe the model's architecture in detail.

**Visual and language encoder** Our visual encoder maps multi-view RGB-D images into a multi-scale 3D scene feature cloud. We use a large-scale pre-trained 2D feature extractor followed by a feature pyramid network [37] to extract multi-scale visual tokens for each camera view. Our input is RGB-D, so each pixel is associated with a depth value. We "lift" the extracted 2D feature vectors to 3D using the pinhole camera equation and the camera intrinsics, based on their average depth. The language encoder featurizes instructions with a large-scale pre-trained language encoder. We use the CLIP ResNet50 [28] visual encoder and language encoders to exploit their common vision-language feature space for interpreting instructions and referential grounding. Our pre-trained visual and language encoders are frozen, not finetuned, during training of Act3D.

**Iterative 3D point sampling and featurization** Our key idea is to estimate high resolution 3D action maps by learning 3D perceptual representations of free space with arbitrary spatial resolution, via recurrent coarse-to-fine 3D point sampling and featurization. 3D point candidates (which we will call ghost points) are sampled, featurized and scored iteratively through relative cross-attention [15] to the physical 3D scene feature cloud, lifted from 2D feature maps of the input image views. We first sample coarsely across the entire workspace, then finely in the vicinity of the ghost point selected as the focus of attention in the previous iteration, as shown in Figure 1. The coarsest ghost points attend to a global coarse scene feature cloud, whereas finer ghost points attend to a local fine scene feature cloud.

**Relative 3D cross-attentions** We featurize each of the 3D ghost points and a parametric query (used to select via inner-product one of the ghost points as the next best end-effector position in the decoder) independently through cross-attentions to the multi-scale 3D scene feature cloud, language tokens, and proprioception. Featurizing ghost points independently, without self-attentions to one another, enables sampling more ghost points at inference time to improve performance, as we show in Section 4. Our cross-attentions use relative 3D position information and are implemented efficiently with rotary positional embeddings [15]. The absolute locations of our 3D points are never used in our featurization, and attentions only depend on the relative locations of two features.

**Decoding actions** We score ghost point tokens via inner product with the parametric query to select one as the next best end-effector position $a_{\text{pos}}$. We then regress the end-effector orientation $a_{\text{rot}}$ and opening $a_{\text{open}}$, as well as whether the motion planner needs to avoid collisions to reach the pose $a_{\text{col}}$, from the last iteration parametric query with a 2-layer multi-layer perceptron (MLP).

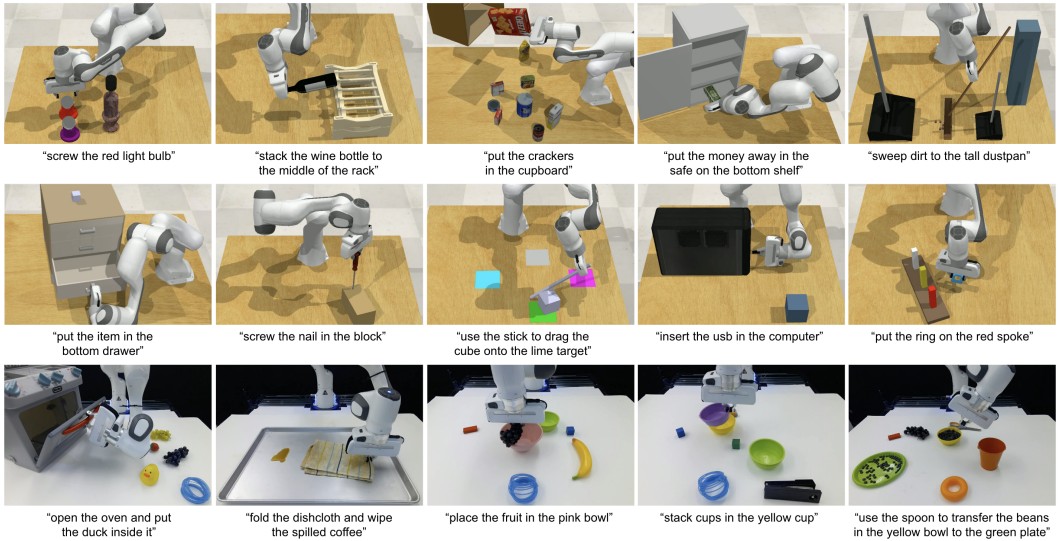

Figure 2: **Tasks.** We conduct experiments on 92 simulated tasks in RLBench [16] (only 10 shown), and 8 real-world tasks (only 5 shown).

**Training** Act3D is trained supervised from input-action tuples from a dataset of manipulation demonstrations. These tuples are composed of RGB-D observations, language goals, and keypose actions $\{(o_1, l_1, k_1), (o_2, l_2, k_2), ...\}$. During training, we randomly sample a tuple and supervise Act3D to predict the keypose action $k$ given the observation and goal $(o, l)$. We supervise position prediction $a_{pos}$ at every round of coarse-to-fine with a softmax cross-entropy loss over ghost points, rotation prediction $a_{rot}$ with a MSE loss on the quaternion prediction, and binary end-effector opening $a_{open}$ and whether the planner needs to avoid collisions $a_{col}$ with binary cross-entropy losses.

**Implementation details** We use three ghost point sampling stages: first uniformly across the entire workspace (roughly 1 meter cube), then uniformly in a 16 centimeter diameter ball, and finally in a 4 centimeter diameter ball. The coarsest ghost points attend to a global coarse scene feature cloud (32x32x$n_{cam}$ coarse visual tokens) whereas finer ghost points attend to a local fine scene feature cloud (the closest 32x32x$n_{cam}$ out of the total 128x128x$n_{cam}$ fine visual tokens). During training, we sample 1000 ghost points in total split equally across the three stages. At inference time, we can trade-off extra prediction precision and task performance for additional compute by sampling more ghost points than the model ever saw at training time ($10,000$ in our experiments). We'll show in ablations in Section 4 that our framework is robust to these hyper-parameters but tying weights across sampling stages and relative 3D cross-attention are both crucial for generalization. We use a batch size 16 on a Nvidia 32GB V100 GPU for 200k steps (one day) for single-task experiments, and a batch size 48 on 8 Nvidia 32GB V100 GPUs for 600K steps (5 days) for language-conditioned multi-task experiments. At test time, we call upon a low-level motion planner to reach predicted keyposes. In simulation, we use native motion planner implementation provided in RLBench, which is a sampling-based BiRRT [38] motion planner powered by Open Motion Planning Library (OMPL) [39] under the hood. For real-world experiments, we use the same BiRRT planner provided by the MoveIt! ROS package [40]. please, see Appendix 7.4 for more details.

## 4   Experiments

We test Act3D in learning from demonstrations single-task and multi-task manipulation policies in simulation and the real world. We conduct our simulated experiments in RLBench [16], an established simulation benchmark for learning manipulation policies, for the sake of reproducibility and benchmarking. Our experiments aim to answer the following questions:

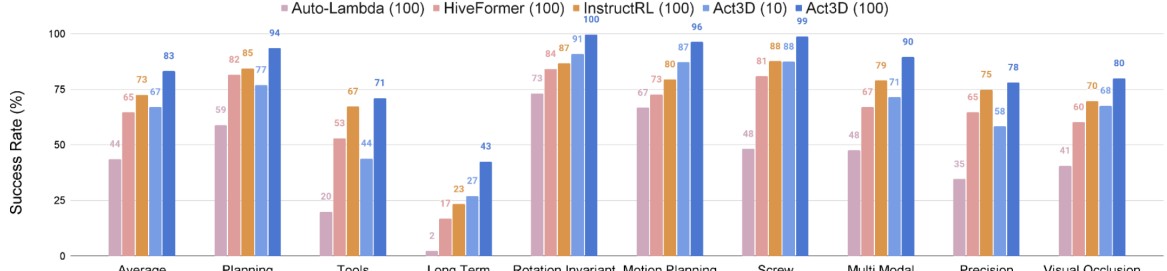

Figure 3: **Single-task performance.** On 74 RLBench tasks across 9 categories, Act3D reaches 83% success rate, an absolute improvement of 10% over InstructRL [3], prior SOTA in this setting.

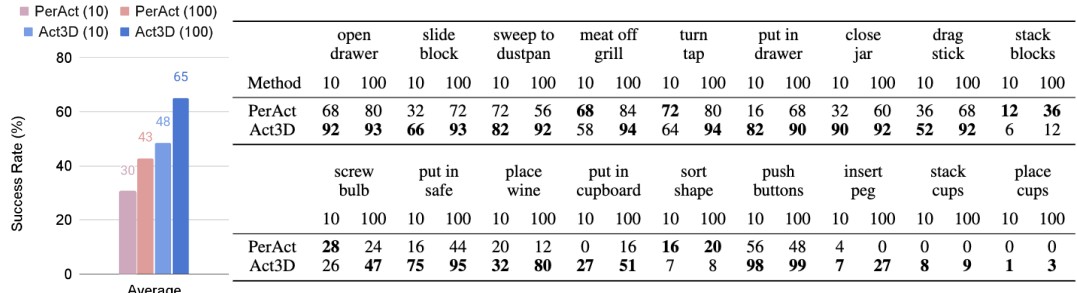

| | open drawer | | slide block | | sweep to dustpan | | meat off grill | | turn tap | | put in drawer | | close jar | | drag stick | | stack blocks | |
|---|---|---|---|---|---|---|---|---|---|---|---|---|---|---|---|---|---|---|
| Method | 10 | 100 | 10 | 100 | 10 | 100 | 10 | 100 | 10 | 100 | 10 | 100 | 10 | 100 | 10 | 100 | 10 | 100 |
| PerAct | 68 | 80 | 32 | 72 | 72 | 56 | **68** | 84 | **72** | 80 | 16 | 68 | 32 | 60 | 36 | 68 | **12** | **36** |
| Act3D | 92 | 93 | 66 | 93 | 82 | 92 | 58 | 94 | 64 | 94 | 82 | 90 | 90 | 92 | 52 | 92 | 6 | 12 |

| | screw bulb | | put in safe | | place wine | | put in cupboard | | sort shape | | push buttons | | insert peg | | stack cups | | place cups | |
|---|---|---|---|---|---|---|---|---|---|---|---|---|---|---|---|---|---|---|
| | 10 | 100 | 10 | 100 | 10 | 100 | 10 | 100 | 10 | 100 | 10 | 100 | 10 | 100 | 10 | 100 | 10 | 100 |
| PerAct | 28 | 24 | 16 | 44 | 20 | 12 | 0 | 16 | **16** | **20** | 56 | 48 | 4 | 0 | 0 | 0 | 0 | 0 |
| Act3D | 26 | **47** | 75 | 95 | 32 | 80 | 27 | 51 | 7 | 8 | 98 | 99 | 7 | 27 | 8 | 9 | 1 | 3 |

Figure 4: **Multi-task performance.** On 18 RLBench tasks with 249 variations, Act3D reaches 65% success rate, an absolute improvement of 22% over PerAct [1], prior SOTA in this setting.

**1.** How does Act3D compare against SOTA 2D multiview and 3D manipulation policies in single-task and multi-task settings with varying number of training demonstrations?

**2.** How does Act3D generalize across camera viewpoints compared to prior 2D multiview policies?

**3.** How do design choices such as relative 3D attention, pre-trained 2D backbones, weight-tied attention layers, and the number of coarse-to-fine sampling stages impact performance?

### 4.1 Evaluation in simulation

**Datasets** We test Act3D in RLbench in two settings: **1. Single-task** manipulation policy learning. We consider 74 tasks grouped into 9 categories proposed by HiveFormer [2]. Each task includes variations which test generalization to novel arrangements of the same training objects. Each method is trained with 100 demonstrations and evaluated on 500 unseen episodes. **2. Multi-task** manipulation policy learning. We consider 18 tasks with 249 variations proposed by PerAct [1]. Each task includes 2-60 variations, which test generalization to new goal configurations that involve novel object colors, shapes, sizes, and categories. This is a more challenging setting. Each method is trained with 100 demonstrations per task split across variations, and evaluated on 500 unseen episodes per task.

**Baselines** We compare Act3D with the following state-of-the-art manipulation policy learning methods: **1.** InstructRL [3], a 2D policy that directly predicts 6 DoF poses from image and language conditioning with a pre-trained vision-and-language backbone. **2.** PerAct [1], a 3D policy that voxelizes the workspace and detects the next best voxel action through global self-attention. **3.** HiveFormer [2] and Auto-$\lambda$ [13], hybrid methods that detect a contact point within an image input, then regress an offset from this contact point. We report numbers from the papers when available.

**Evaluation metric** We evaluate policies by task completion success rate, the proportion of execution trajectories that lead to goal conditions specified in language instructions.

**Single-task and multi-task manipulation results**    We show single-task quantitative results of our model and baselines in Figure 3. Act3D **reaches 83% success rate, an absolute improvement of 10% over InstructRL [3], prior SOTA in this setting**, and consistently outperforms it across all 9 categories of tasks. With only 10 demonstrations per task, Act3D is competitive with prior SOTA using 100 demonstrations per task. Act3D outperforms 2D methods of InstructRL and Hiveformer because it reasons directly in 3D. For the same reason, it generalizes much better than them to novel camera placements, as we show in Table 3.

We show multi-task quantitative results of our model and PerAct in Figure 4. Act3D reaches 65% success rate, an absolute improvement of 22% over PerAct, prior SOTA in this setting, consistently outperforming it across most tasks. **With only 10 demonstrations per task, Act3D outperforms PerAct using 100 demonstrations per task.** Note that Act3D also uses less than a third of PerAct's training computation budget: PerAct was trained for 16 days on 8 Nvidia V100 GPUs while we train for 5 days on the same hardware. Act3D outperforms PerAct because its coarse-to-fine relative attention based 3D featurization of the 3D workspace is more effective than the perceiver's latent bottleneck attention in generating spatially disentangled features.

## 4.2    Evaluation in real-world

In our real-world setup, we conduct experiments with a Franka Emika Panda robot and a single Azure Kinect RGB-D sensor. We consider 8 tasks (Figure 2) that involve interactions with multiple types of objects, spanning liquid, articulated objects, and deformable objects. For each task, we collected 10 to 15 kinesthetic demonstrations and trained a languaged-conditioned multi-task model with all of them. We report the success rate on 10 episodes per task in Table 1. Act3D can capture semantic knowledge in demonstration well and performs reasonably well on all tasks, even with a single camera input. One major failure case comes from noisy depth sensing: when the depth image is not accurate, the selected point results in imprecise action prediction. Leveraging multi-view input for

| Task | # Train | Success |
|---|---|---|
| reach target | 10 | 10/10 |
| duck in oven | 15 | 6/10 |
| wipe coffee | 15 | 7/10 |
| fruits in bowl | 10 | 8/10 |
| stack cups | 15 | 6/10 |
| transfer beans | 15 | 5/10 |
| press handsan | 10 | 10/10 |
| uncrew cap | 10 | 8/10 |

Table 1: Real-world tasks.

error correction could improve this, and we leave this for future work. For videos of the robot executing the tasks, please see our project website.

## 4.3    Ablations

We ablate the impact of our design choices in Table 3. We perform most ablations in the single-task setting on 5 tasks: pick cup, put knife on chopping board, put money in safe, slide block to target, take umbrella out of stand. We ablate the choice of pre-trained 2D backbone in the multi-task setting with all 18 tasks.

**Generalization across camera viewpoints:**    We vary camera viewpoints at test time for both Act3D and HiveFormer [2]. The success rate drops to 20.4% for HiveFormer, a relative 77% drop, while Act3D achieves 74.2% success rate, a 24% relative drop. This shows detecting actions in 3D makes Act3D more robust to camera viewpoint changes than multiview 2D methods that regress offsets.

**Weight-tying and coarse-to-fine sampling:**    All 3 stages of coarse-to-fine sampling are necessary: a model with only 2 stages of sampling and regressing an offset from the position selected at the second stage suffers a 4.5% performance drop. Tying weights across stages and relative 3D positional embeddings are both crucial; we observed severe overfitting without, reflected in respective 17.5% and 42.7% performance drops. Fine ghost point sampling stages should attend to local fine visual features with precise positions: all stages attending to global coarse features leads to a 8.3% performance drop. Act3D can effectively trade off inference computation for performance:

Table 2: **Ablations.**

|  |  | Average success rate in single-task setting (5 tasks) |
| --- | --- | --- |
| Core design choices | Full Act3D | **98.1** |
|  | Only 2 stages of coarse-to-fine sampling | 93.6 |
|  | No weight tying across stages | 80.6 |
|  | Absolute 3D positional embeddings | 55.4 |
|  | Attention to only global coarse visual features | 89.8 |
|  | Only 1000 ghost points at inference time | 93.2 |
| Viewpoint changes | Act3D | **74.2** |
|  | HiveFormer | 20.4 |
|  |  | Multi-task setting (18 tasks) |
| Backbone | CLIP ResNet50 backbone | **65.1** |
|  | ImageNet ResNet50 backbone | 53.4 |

sampling 10,000 ghost points, instead of the 1,000 the model was trained with, boosts performance by 4.9%.

**Pre-training 2D features:** We investigate the effect of the pre-trained 2D backbone in the multi-task setting where language instructions are most needed. A ResNet50 [28] backbone pre-trained with CLIP improves success rate by 8.7% over a ResNet50 backbone pre-trained on ImageNet.

For additional ablations regarding augmentations and sensitivity to hyperparameters, please see the Appendix section 7.6. We found Random crops of RGB-D images to boost performance but yaw rotation perturbations did not help. The model is robust to variations in hyperparameters such as the diameter of ghost point sampling balls or the number of points sampled during training.

### 4.4 Limitations and future work

Our framework currently has the following limitations: **1.** Act3D is limited by the motion planner used to connect predicted keyposes with straight trajectory segments. It does not handle manipulation of articulated object well, such as opening/closing doors, fridges, and ovens, where robot trajectories cannot be well approximated by few line segments.**2.** Act3D does not utilize any decomposition of tasks into subtasks. A hierarchical framework that would predict language subgoals for subtasks [41, 42, 43] and feed those to our language-conditioned policy would allow better reusability of skills across tasks. Addressing these limitations is a direct avenue for future work.

## 5 Conclusion

We presented Act3D, a language-conditioned policy transformer that predicts continuous resolution 3D action maps for multi-task robot manipulation. Act3D represents the scene using a continuous resolution 3D feature map, obtained by coarse-to-fine 3D point sampling and attention-based featurization. Act3D sets a new state-of-the-art in RLBench, an established robot manipulation benchmark, and solves diverse manipulation tasks in the real world from a single RGB-D camera view and a handful of demonstrations. Our ablations quantified the contribution of relative 3D attentions, 2D feature pre-training, and weight tying during coarse-to-fine iterations.

## 6 Acknowledgements

This work is supported by Sony AI, NSF award No 1849287, DARPA Machine Common Sense, an Amazon faculty award, and an NSF CAREER award.

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

# 7 Appendix

## 7.1 Real-world Setup

Our real-robot setup contains a Franka Panda robotic arm equipped with a parallel jaw gripper, as shown in Figure 5. We get RGB-D input from a single Azure Kinect sensor at a front view at 30Hz. The image input is of resolution $1280 \times 720$, we crop and downsample it to $256 \times 256$. We calibrate the extrinsics of the camera with respect to the robot base using the `easy_handeye`[1] ROS package. We extract keyposes from demonstrations in the same was as in simulation. Our real-world multi-task policy is trained on 4 V100 GPUs for 3 days, and we run inference on a desktop with a single RTX4090 GPU. For robot control, we use the open-source `frankapy`[2] package to send real-time position-control commands to the robot.

## 7.2 RLBench Simulation Setup

To ensure fair comparison with prior work, we use $n_{\text{cam}} \in \{3, 4\}$ cameras for simulated experiments depending on the evaluation setting. In our single-task evaluation setting first proposed by HiveFormer [2], we use the same 3 cameras they do $\{O_{\text{left}}, O_{\text{right}}, O_{\text{wrist}}\}$. In our multi-task evaluation setting first proposed by PerAct [1], we use the same 4 cameras they do $\{O_{\text{front}}, O_{\text{left}}, O_{\text{right}}, O_{\text{wrist}}\}$.

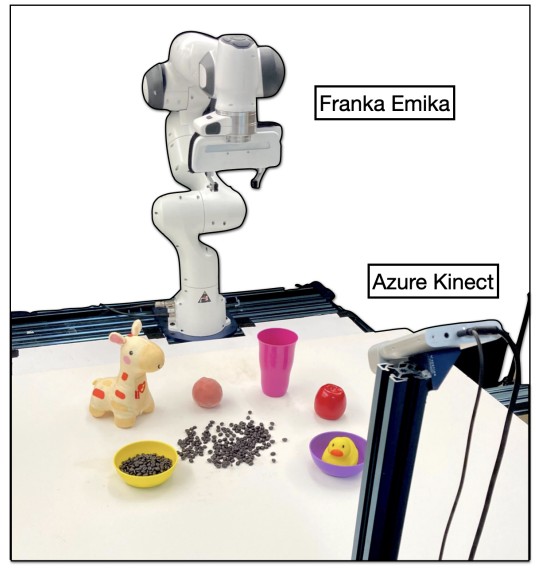

Figure 5: **Real-world setup.**

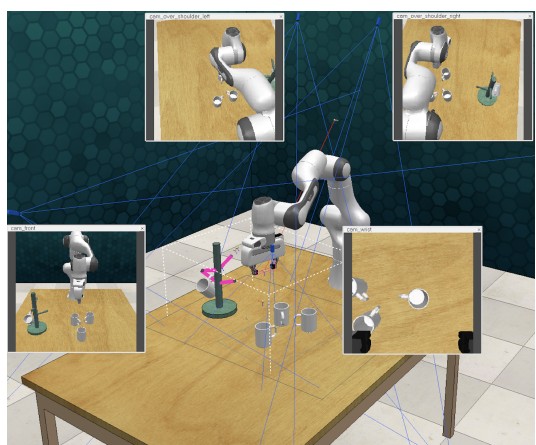

Figure 6: **RLbench simulation setup.**

[1] https://github.com/IFL-CAMP/easy_handeye
[2] https://github.com/iamlab-cmu/frankapy

## 7.3 RLBench Tasks

| Task | Variation Type | # of Variations | Avg. Keyposes | Language Template |
|------|---------------|-----------------|---------------|-------------------|
| `open drawer` | placement | 3 | 3.0 | "open the ___ drawer" |
| `slide block` | color | 4 | 4.7 | "slide the block to ___ target" |
| `sweep to dustpan` | size | 2 | 4.6 | "sweep dirt to the ___ dustpan" |
| `meat off grill` | category | 2 | 5.0 | "take the ___ off the grill" |
| `turn tap` | placement | 2 | 2.0 | "turn ___ tap" |
| `put in drawer` | placement | 3 | 12.0 | "put the item in the ___ drawer" |
| `close jar` | color | 20 | 6.0 | "close the ___ jar" |
| `drag stick` | color | 20 | 6.0 | "use the stick to drag the cube onto the ___ target" |
| `stack blocks` | color, count | 60 | 14.6 | "stack ___ ___ blocks" |
| `screw bulb` | color | 20 | 7.0 | "screw in the ___ light bulb" |
| `put in safe` | placement | 3 | 5.0 | "put the money away in the safe on the ___ shelf" |
| `place wine` | placement | 3 | 5.0 | "stack the wine bottle to the ___ of the rack" |
| `put in cupboard` | category | 9 | 5.0 | "put the ___ in the cupboard" |
| `sort shape` | shape | 5 | 5.0 | "put the ___ in the shape sorter" |
| `push buttons` | color | 50 | 3.8 | "push the ___ button, [then the ___ button]" |
| `insert peg` | color | 20 | 5.0 | "put the ring on the ___ spoke" |
| `stack cups` | color | 20 | 10.0 | "stack the other cups on top of the ___ cup" |
| `place cups` | count | 3 | 11.5 | "place ___ cups on the cup holder" |

Figure 7: **PerAct [1] tasks.** We adopt the multi-task multi-variation setting from PerAct [1] with 18 tasks and 249 unique variations across object placement, color, size, category, count, and shape.

We adapt the single-task setting of HiveFormer [2] with 74 tasks grouped into 9 categories according to their key challenges. The 9 task groups are defined as follows:

- The **Planning** group contains tasks with multiple sub-goals (e.g. picking a basket ball and then throwing the ball). The included tasks are: basketball in hoop, put rubbish in bin, meat off grill, meat on grill, change channel, tv on, tower3, push buttons, stack wine.

- The **Tools** group is a special case of planning where a robot must grasp an object to interact with the target object. The included tasks are: slide block to target, reach and drag, take frame off hanger, water plants, hang frame on hanger, scoop with spatula, place hanger on rack, move hanger, sweep to dustpan, take plate off colored dish rack, screw nail.

- The **Long term** group requires more than 10 macro-steps to be completed. The included tasks are: wipe desk, stack blocks, take shoes out of box, slide cabinet open and place cups.

- The **Rotation-invariant** group can be solved without changes in the gripper rotation. The included tasks are: reach target, push button, lamp on, lamp off, push buttons, pick and lift, take lid off saucepan.

- The **Motion planner** group requires precise grasping. As observed in [81] such tasks often fail due to the motion planner. The included tasks are: toilet seat down, close laptop lid, open box, open drawer, close drawer, close box, phone on base, toilet seat up, put books on bookshelf.

- The **Multimodal** group can have multiple possible trajectories to solve a task due to a large affordance area of the target object (e.g. the edge of a cup). The included tasks are: pick up cup, turn tap, lift numbered block, beat the buzz, stack cups.

- The **Precision** group involves precise object manipulation. The included tasks are: take usb out of computer, play jenga, insert onto square peg, take umbrella out of umbrella stand, insert usb in computer, straighten rope, pick and lift small, put knife on chopping board, place shape in shape sorter, take toilet roll off stand, put umbrella in umbrella stand, setup checkers.

- The **Screw** group requires screwing an object. The included tasks are: turn oven on, change clock, open window, open wine bottle.

- The **Visual Occlusion** group involves tasks with large objects and thus there are occlusions from certain views. The included tasks are: close microwave, close fridge, close grill, open

grill, unplug charger, press switch, take money out safe, open microwave, put money in safe, open door, close door, open fridge, open oven, plug charger in power supply

## 7.4 Further Architecture Details

**Relative 3D cross-attentions**   We featurize each of the 3D ghost points and a parametric query (used to select via inner-product one of the ghost points as the next best end-effector position in the decoder) independently through cross-attentions to the multi-scale 3D scene feature cloud, language tokens, and proprioception. Featurizing ghost points independently, without self-attentions to one another, enables sampling more ghost points at inference time to improve performance, as we show in Section 4. Our cross-attentions use relative 3D position information and are implemented efficiently with rotary positional embeddings [15].

Given a point $\mathbf{p} = (x, y, z) \in \mathbb{R}^3$ and its feature $\mathbf{x} \in \mathbb{R}^d$, the rotary position encoding function $\mathbf{PE}$ is defined as:

$$\mathbf{PE}(\mathbf{p}, \mathbf{x}) = \mathbf{M}(\mathbf{p})\mathbf{x} = \begin{bmatrix} \mathbf{M}_1 & & \\ & \ddots & \\ & & \mathbf{M}_{d/6} \end{bmatrix} \mathbf{x} \tag{1}$$

$$\mathbf{M}_k = \begin{bmatrix} \cos x\theta_k & -\sin x\theta_k & 0 & 0 & 0 & 0 \\ \sin x\theta_k & \cos x\theta_k & 0 & 0 & 0 & 0 \\ 0 & 0 & \cos y\theta_k & -\sin y\theta_k & 0 & 0 \\ 0 & 0 & \sin y\theta_k & \cos y\theta_k & 0 & 0 \\ 0 & 0 & 0 & 0 & \cos z\theta_k & -\sin z\theta_k \\ 0 & 0 & 0 & 0 & \sin z\theta_k & \cos z\theta_k \end{bmatrix} \tag{2}$$

where $\theta_k = \frac{1}{10000^{6(k-1)/d}}$, $k \in \{1, .., d/6\}$. The dot product of two positionally encoded features is

$$\mathbf{PE}(\mathbf{p}_i, \mathbf{x}_i)^T \mathbf{PE}(\mathbf{p}_j, \mathbf{x}_j) = \mathbf{x}_i^T \mathbf{M}(\mathbf{p}_i)^T \mathbf{M}(\mathbf{p}_j)\mathbf{x}_j = \mathbf{x}_i^T \mathbf{M}(\mathbf{p}_j - \mathbf{p}_i)\mathbf{x}_j \tag{3}$$

which depends only on the relative positions of points $\mathbf{p}_i$ and $\mathbf{p}_j$.

We extract two feature maps per 256x256 input image view: 32x32 coarse visual tokens and 128x128 fine visual tokens. We use three ghost point sampling stages: first uniformly across the entire workspace (roughly 1 meter cube), then uniformly in a 16 centimeter diameter ball, and finally in a 4 centimeter diameter ball. The coarsest ghost points attend to a global coarse scene feature cloud (32x32x$n_{\mathrm{cam}}$ coarse visual tokens) whereas finer ghost points attend to a local fine scene feature cloud (the closest 32x32x$n_{\mathrm{cam}}$ out of the total 128x128x$n_{\mathrm{cam}}$ fine visual tokens). During training, we sample 1000 ghost points in total split equally across the three stages. At inference time, we can trade-off extra prediction precision and task performance for additional compute by sampling more ghost points than the model ever saw at training time ($10,000$ in our experiments). We'll show in ablations in Section 4 that our framework is robust to these hyper-parameters but tying weights across sampling stages and relative 3D cross-attention are both crucial for generalization. We use 2 layers of cross-attention and an embedding size 60 for single-task experiments and 120 for multi-task experiments. Training samples are augmented with random crops of RGB-D images and $\pm 45.0$ yaw rotation perturbations (only in the real world as this degrades performance in simulation as we'll show in Section 4). The cropping operation is performed on aligned RGB and depth frames together, thus maintain pixel-level correspondence. We use a batch size 16 on a Nvidia 32GB V100 GPU for 200k steps (one day) for single-task experiments, and a batch size 48 on 8 Nvidia 32GB V100 GPUs for 600K steps (5 days) for language-conditioned multi-task experiments. At test time, we call a low-level motion planner to reach predicted keyposes. In simulation, we use native motion planner implementation provided in RLBench, which is a sampling-based BiRRT [38] motion planner powered by Open Motion Planning Library (OMPL) [39] under the hood. For real-world experiments, we use the same BiRRT planner provided by the MoveIt! ROS package [40].

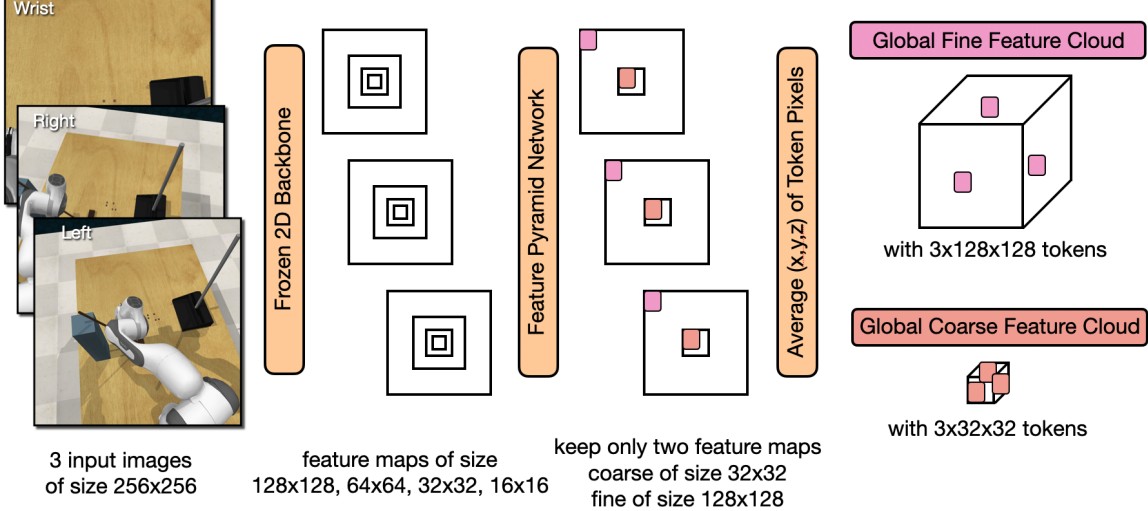

Figure 8: **Scene Feature Cloud Generation**. We encode each image independently with a pre-trained and frozen vision backbone to get multi-scale feature maps, pass these feature maps through a feature pyramid network and retain only two: a coarse feature map (at a granularity that lets ghost points attend to all tokens within GPU memory) and a fine feature map (as spatially precise as afforded by input images and the backbone). We lift visual tokens from these two feature maps for each image to 3D scene feature clouds by averaging the positions of pixels in each 2D visual token.

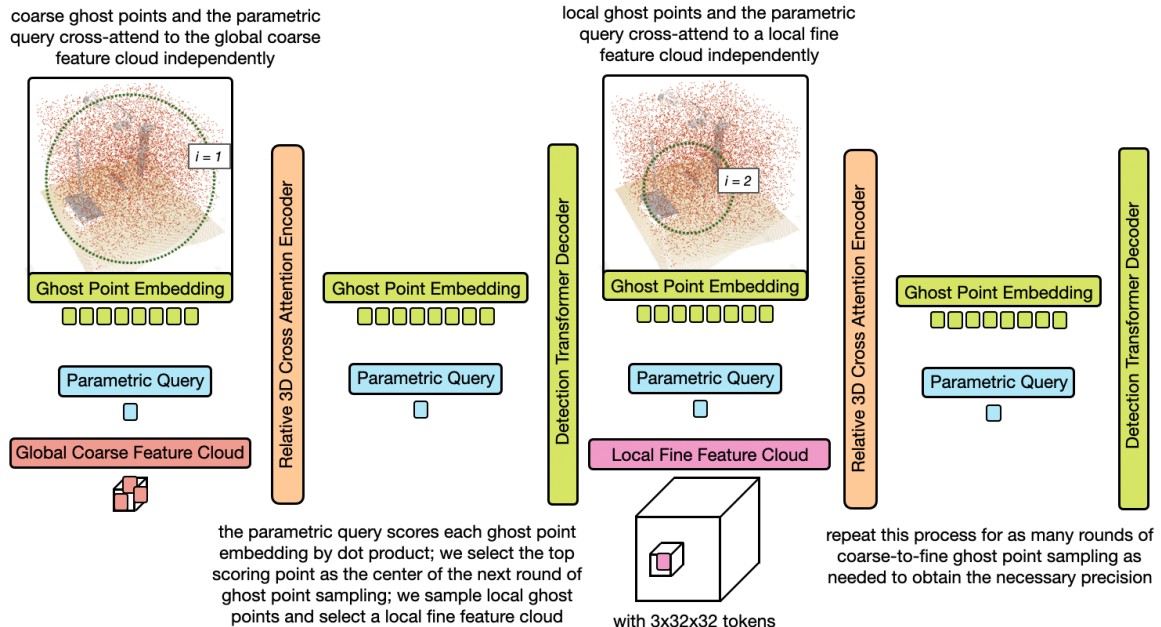

Figure 9: **Iterative Ghost Point Sampling, Featurization, and Selection**.

## 7.5 High Precision Experiments

In this section, we further investigate the ability of Act3D to improve over existing 3D methods that voxelize the workspace for high-precision tasks. We compare two variants of Act3D against PerAct [1] on three high-precision tasks in success rate. The first Act3D variant is the standard architecture used in the remainder of our experiments operating on 256x256 input image views; the second operates on higher resolution 512x512 input image views, from which it extracts four times as many visual tokens with more precise 3D positions. This further tests the ability of Act3D to provide high precision by processing higher-resolution RGB-D views at the cost of extra compute.

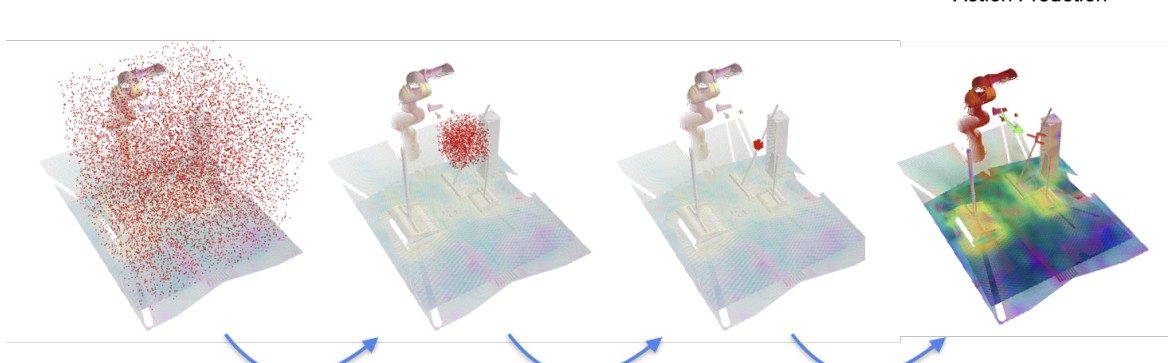

Figure 10: **Iterative Ghost Point Sampling, Featurization, and Selection**.

| Method | insert peg | sort shape | screw nail |
|---|---|---|---|
| PerAct | 16 | 31 | 12 |
| Act3D (256x256) | 29 | 34 | 31 |
| Act3D (512x512) | **47** | **43** | **55** |

Act3D improves over PerAct on high precision tasks and can further benefit from higher resolution RGB-D images, at the cost of extra compute.

## 7.6 Further ablations

**Augmentations:** Random crops of RGB-D images boost success rate by 6.5%, but yaw rotation perturbations drop it by 11.9%. This is in line with PerAct [1] results in RLBench.

**Hyperparameter sensitivity:** Act3D is robust to variations in hyperparameters. Doubling the diameter of ghost point sampling balls from (16 cm, 4 cm) to (32 cm, 8 cm) drops success rate by 1.5% and halving it to (8 cm, 2 cm) by 6.9%. Halving the total number of ghost points sampled from 1,000 to 500 drops success rate by 2.3% whereas doubling it to 2,000 increases success rate by 0.3%. We use 1,000 ghost points in our experiments to allow training with a single GPU per task.

Table 3: **Ablations.**

| | Model | Average success rate in single-task setting (5 tasks) |
|---|---|---|
| Core design choices | Best Act3D model (evaluated in Fig. 3) | **98.1** |
| | Only 2 stages of coarse-to-fine sampling: full workspace, 16 cm ball, regress an offset | 93.6 |
| | No weight tying across stages | 80.6 |
| | Absolute 3D positional embeddings | 55.4 |
| | Attention to only global coarse visual features | 89.8 |
| | Only 1000 ghost points at inference time | 93.2 |
| Viewpoint changes | Best Act3D model (evaluated in Fig. 3) | **74.2** |
| | HiveFormer | 20.4 |
| Augmentations | No image augmentations | **91.6** |
| | With rotation augmentations | 86.2 |
| Hyperparameter sensitivity | Double sampling ball diameters: 32 cm and 8 cm | 96.6 |
| | Halve sampling ball diameters: 8 cm and 2 cm | 91.2 |
| | 500 ghost points at training time | 95.8 |
| | 2000 ghost points at training time (need 2 GPUs) | **98.4** |

| | | Multi-task setting (18 tasks) |
|---|---|---|
| Backbone | CLIP ResNet50 backbone | **65.1** |
| | ImageNet ResNet50 backbone | 53.4 |
| | No backbone (raw RGB) | 45.2 |

