# OpenReview forum: "Act3D: 3D Feature Field Transformers for Multi-Task Robotic Manipulation"
_robot-learning.org/CoRL/2023/Conference — CoRL 2023 Poster_

### Official Review · Reviewer_C6Eb · 2023-07-02

**Confidence:** 4
**Originality:** Very Good
**Technical Quality:** Good
**Clarity Of Presentation:** Good
**Impact:** 4

**Recommendation:**

Weak Accept: I recommend accepting the paper, but will not argue for my recommendation if the majority of other reviewers have a different opinion.

**Review:**

Quality, clarity, originality and significance of this work:
The major novelty is leverage ghost points instead of voxels (in PerAct) representing action candidates to reduce computation burden and improve resolution. Moreover, this work combines multiple idea in the literature (image and language pre-training, relative position embedding, key-point estimation, coarse-to-fine evaluation). Together, these novelties contribute to manipulation policy learning research.
Overall the structure of the paper is well-organized.
Strengths:
Point-cloud based transformer and pre-trained vision model effectively improves task success rate and reduces computation overhead.
Weaknesses:
Insufficient description for “Detection Transformer decoder”.
As described in the title, the method supports high-resolution. Nevertheless, the method does not achieves signification improvement in precision tasks, i.g., Precision in Figure. 3 and “sort shape”, “insert peg” in Figure. 4.

**Quality Of The Limitations Section:**

Limitations are addressed clearly

**Questions For Rebuttal:**

1. The major novelty ghost point evaluation, i.e., Detection Transformer decoder in line 141 is poorly presented. This is main learnable neural network in the method as well. But its architecture is unclear. Please add more description.
2. Another concern is the method supports high-resolution, but some of the experiments (as mentioned in weaknesses) does not indicate the benefit of the high-resolution. It would be nice to have experiment show the benefit of high-resolution.
3. In the ablations (Table 2), when the image backbone is removed from the method, performance dropped to 45.2%, which is similar to that of PerAct (43%). Does this mean the task successes rate advantage mainly comes from the pre-trained vision model?
4. It is also worth mention how the ghost point is sampled.
5. Minor: some figures are not properly referenced, for example Fig.4 is not mentioned in Multi-task manipulation results.

**Robotics Focus:**

Sufficient demonstration on hardware

**Summary Of Paper:**

This paper proposes a novel manipulation policy transformer that leverages multi-view camera information and selects the end-effector action (SE(3) position and some other actions). The policy first extracts features from image observation and language goal with pre-trained models, projects the features to the scene point cloud, then evaluates sampled ghost points in the X-Y-Z- action space using a transformer in a coarse-to-fine style. Lastly, regress the other actions. Experiments indicate the transformer outperforms baselines in both single-task and multi-task in RLBench. Finally, real-world experiments verified the capability of the transformer in practice.

**Summary Of Recommendation:**

This paper proposes a manipulation policy transformer to address two issues in previous SOTA method PerAct, which are computation burden and action descretization granularity. The method effectively achieves better performance with less computation. However, the advantage of high-resolution is not explicitly demonstrated.

Updates after rebuttal:
I appreciate authors' explanations and additional experiments. I keep my current rating (weak accept). These new experiments support the high resolution contributes to the task success rate. However, it remains a question that what could improve the task success rate from ~65%  to 100%.

---

### Official Review · Reviewer_fEwK · 2023-07-19

**Confidence:** 4
**Originality:** Very Good
**Technical Quality:** Very Good
**Clarity Of Presentation:** Good
**Impact:** 3

**Recommendation:**

Strong Accept: I recommend accepting the paper and will argue for my recommendation even if other reviewers hold a different opinion.

**Review:**

**Strengths:**
- The contribution of the paper - multi-task robot manipulation policy learning in 3D - addresses an important challenge of spatial precision required in manipulation tasks.
- While some previous methods have addressed this challenge, this paper tackles the problem with lesser computational requirements and fewer amounts of data.
- The method is more general by predicting the key poses of the end-effector toward performing a task. It can easily plug the advancements in motion planning and trajectory optimization research areas into the pipeline.
- Reported results substantiate the contributions listed.

**Weaknesses:**
- In the current version of the paper, 3D Feature cloud generation that lifts the 2D visual tokens into 3D needs to be clarified. Particularly the usage of the features from the feature pyramid and if multiple 3D points will hold the same features in this process. An additional diagram to illustrate this will be beneficial, as this is a crucial part of the proposed method.
- In the current version of the paper, the description of augmentation with random crops needs to be clarified, as the association between RGB and depth will likely be lost.
- As the robot is part of the observation space, the developed method is catered to the setup and the specific robot.

**Quality Of The Limitations Section:**

Limitations are addressed clearly

**Questions For Rebuttal:**

- Can the authors elaborate on 3D feature cloud generation in more detail?
- Can the authors elaborate on the augmentation with random crops?
- Can the authors explain the significance of the parametric query? And how this changes for every iteration?

**Robotics Focus:**

Sufficient demonstration on hardware

**Summary Of Paper:**

This paper introduces Act3D, a manipulation policy Transformer designed to address the computational challenges associated with 3D representations. It proposes a novel approach by casting 6-DoF key pose prediction as 3D detection with adaptive spatial computation. It takes in multi-view RGB-D observations, language instruction, and proprioception data of the current end-effector as inputs to generate action in the form of end-effector pose, gripper state, and a binary describing the collision allowance. From the input RGB-D observations, it generates 3D feature clouds and employs iterative coarse-to-fine sampling of 3D point grids in free space. These grids are then featurized using relative spatial attention to the physical feature cloud, allowing Act3D to select the optimal end-effector position. The results in the paper indicate that Act3D achieves improvement over the previous SOTA with only one-third of the computational resources compared to the previous SOTA method.

**Summary Of Recommendation:**

I believe the work would be valuable to the community. I would consider increasing my recommendation if the questions for the rebuttal phase are addressed.

**Post-rebuttal update:**
The authors have addressed my concerns satisfactorily. I appreciate the significant improvement in the quality of the manuscript. My recommendation to accept this paper remains.

---

### Official Review · Reviewer_hSNU · 2023-07-19

**Confidence:** 4
**Originality:** Very Good
**Technical Quality:** Very Good
**Clarity Of Presentation:** Very Good
**Impact:** 3

**Recommendation:**

Weak Accept: I recommend accepting the paper, but will not argue for my recommendation if the majority of other reviewers have a different opinion.

**Review:**

Pros:
- Solves a real problem in keypoint-based manipulation
- Strong results on a standard benchmark for the task
- The paper includes convincing RLBench experimental results showing significant improvements over prior single-task and multi-task keypoint based manipulation policies. Real-world experiments showing success rates across a set of tasks are also included, showing the ability to learn from limited data in real world.

Cons (not really cons, but thought I’d mention):
- the main drawback is the problem formulation and not the paper - keypoint based manipulation is not able to solve dynamic dexterous manipulation tasks. However, the paper should not be penalized for it, since it is still very useful for a large amount of real world tasks, and is able to learn from limited data - a key requirement for real-world applicability.
- Requires a calibrated setup with known camera poses. Again, this is currently a standard assumption in related work and still practical in uses like industrial work cells.

**Quality Of The Limitations Section:**

Limitations are addressed clearly

**Questions For Rebuttal:**

generally the paper is good enough as-is and I would recommend acceptance without any additional work. It is easy to read, conveys a clear message, and is not unnecessarily overstuffed.

**Robotics Focus:**

Sufficient demonstration on hardware

**Summary Of Paper:**


Recently keypoint-based manipulation has seen increasing attention in robotics, where a neural network (typically a transformer) is trained to predict a sequence of end-effector keypoints to be used as motion planner goals. Prior approaches suffer from poor spatial resolution or errors due to the scene representation to the transformer (e.g. voxels or images), of which voxels additionally impose a n^3 runtime. Ideally, the representation used should be sparse, and this paper suggests a featurized pointcloud. Unlike voxels, a point-cloud is only defined on surfaces, but end-effector poses are almost always in free space. Thus, a point cloud does not offer input tokens for the transformer at positions corresponding to output poses. This paper iteratively samples, computes and refines refines a set of featurized ghost points in free space to enable output of end-effector poses in continuous free space, not suffering from the limited resolution or view-dependency of prior methods

**Summary Of Recommendation:**

It is a good paper that proposed a useful method and is substantiated by sufficient experiments and ablations.

---

### Official Review · Reviewer_4R1b · 2023-07-24

**Confidence:** 3
**Originality:** Very Good
**Technical Quality:** Very Good
**Clarity Of Presentation:** Good
**Impact:** 3

**Recommendation:**

Weak Accept: I recommend accepting the paper, but will not argue for my recommendation if the majority of other reviewers have a different opinion.

**Review:**

The paper studies an important problem in robot manipulation and proposes a novel solution to address it. However, the manuscript requires significant improvement in writing. In its current form, it fails to motivate the problem correctly and does not convey the crucial ideas about the approach comprehensively. It is unclear what are the main technical contributions of the work. Hence, I am inclined to reject the manuscript.

More detailed feedback is as follows:
- Please add examples of "... scene features..." referred to in line 28 for more clarity.
- It is unclear what is meant by the phrases "... predicted on 3D points in free space..." and "...3D physical scene points" in lines 31-32. Consider rephrasing the sentence for better readability.
- Consider adding examples and more context with the phrase "2D backbones and back-project to 3D" in lines 35-37. Please rephrase line 37 for better readability.
- It is difficult to follow the argument being presented in lines 39-40. Please rephrase.
- In line 103 it is stated that Act3D predicts s 6-Dof end-effctor pose at each time step. However, in line 111 it is said that "Following prior work [11, 1, 2, 3], instead of predicting an end-effector pose at each timestep, we extract a set of keyposes that capture bottleneck end-effector poses in a demonstration", which seems to contradict the previous statement. Please clarify.
- Please add further details on the motion planner used in the method.
- Consider describing how the "2D visual tokens are lifted to 3D by interpolating depth values" in the text.
- Consider adding more details on how the end-effector orientation a_{ori} and opening/closing action a_{open} are predicted.
- In the included video, excessive end-effector rotations were observed in some simulated experiments, e.g., "screw the red light bulb", and "put the item in the bottom drawer". Please discuss the observation.
- In the single-task and multi-task manipulation results in sections (lines 193-209), consider discussing which components of the proposed approach helped it to outperform SoTA methods for a better emphasis on the merits of the approach.
- Consider adding evaluations to a more diverse set of baselines (e.g. [1], [2], [3]) to clearly demonstrate the effectiveness of the method.
- Typographical:
  - map -> maps [line 119]
  - No line number is available for the paragraph between lines 117 and 118. Please check the paper format.
  - Consider describing a range of suffix 'k' in the equation between lines 139-140.  Also, please add an equation number for easy reference.
  - Rephrase "...which test generalization to test goal..." in line 200.
  - Consider replacing the phrase "... robust to hyperparameters" --> "robust to variations in hyperparameters".

References:
- [1] Brohan, Anthony, et al. "Rt-1: Robotics transformer for real-world control at scale." arXiv preprint arXiv:2212.06817 (2022).
- [2] Driess, Danny, et al. "Palm-e: An embodied multimodal language model." arXiv preprint arXiv:2303.03378 (2023).
- [3] Lin, Kevin, et al. "Text2motion: From natural language instructions to feasible plans." arXiv preprint arXiv:2303.12153 (2023).

**Quality Of The Limitations Section:**

Limitations are addressed clearly

**Questions For Rebuttal:**

1. In the included video, excessive end-effector rotations were observed in some simulated experiments, e.g., "screw the red light bulb", and "put the item in the bottom drawer". Please discuss the observation.
2. Please clarify the technical contributions of the work.

**Robotics Focus:**

Sufficient demonstration on hardware

**Summary Of Paper:**

The paper introduces Act3D, a novel manipulation policy transformer, for performing 6D robot manipulation tasks given natural language description and multi-view RGB-D images of the workspace. The method uses a coarse-to-fine sampling method to create a 3D feature cloud of candidate end-effector poses for the task, which is later queried to predict the keyposes required to complete the task. A motion planner is used to find collision-free motion plans to connect these keyposes. The proposed method outperforms the state-of-the-art policy transformer methods on the RLBench robot manipulation benchmark for performing single and multi-task manipulation tasks.

**Summary Of Recommendation:**

The manuscript in its current form requires significant improvements. It fails to motivate the problem correctly and does not convey the crucial ideas about the approach comprehensively. It is unclear what are the main technical contributions of the work. Hence, I am inclined to reject the manuscript.

**Post-rebuttal update:**
The authors have addressed my concerns satisfactorily. I appreciate the significant improvement in the quality of the manuscript. I have updated my recommendation accordingly.

---

### Author Response · Authors · 2023-08-15
**General Response**

We thank the reviewers for their constructive feedback.
The reviewers appreciate the method presented in the paper, they argue it tackles an important problem (4R1b,hSNU,fEwK), agree on its novelty (hSNU,fEwK,C6Eb) and on its good empirical performance (hSNU,fEwK,C6Eb).

We revised the paper draft in blue to address reviewers’ concerns, and summarize the major changes below:
- Rewriting the introduction and clarifying the method’s contributions (4R1b)
- Additional figures and  clearer writing of technical details to better explain the method (fEwK, C6Eb)
- Additional experiments on high precision manipulation tasks (C6Eb)
- Incorporating all suggestions of reviewers on paper writing, discussion of experimental results, figures and technical detail presentation. (4R1b, hSNU, fEwK, C6Eb)

We address each reviewer’s concerns in their respective rebuttal response.

---

### Decision · Program_Chairs · 2023-08-30

**Decision:**

Accept (Poster)

**Comment:**

The authors present Act3D, a method for 6D language-conditioned robot manipulation from multi-view RGB images. Compared to similar sparse-trajectory (open-loop) style approaches, the author’s course-to-fine method is able to infer actions continuously on SE(3), overcoming the inherent limitation of discretization of other approaches.

Owing to strong real-world results on benchmarks that matter, and a clear and concise manuscript (after revisions), reviewer sentiment was uniformly in favor of acceptance. Likewise, the authors presented a strong rebuttal, with additional experiments, a re-written introduction and clarifications on points raised by the reviewers.

A number of reviewers raised a weakness inherent to this style of approach; that it is not appropriate for high-rate dexterous tasks or (as the empirical results suggest) does not outperform baselines on tasks with high precision. However, such tasks are likely outside the scope of this work and the strong results on the evaluated tasks against fair baselines was deemed sufficient for acceptance, and generally for this paper to be of interest to the community.